# Predictors of low birthweight and comparisons of newborn birthweights among different groups of maternal factors at Rev. John Chilembwe Hospital in Phalombe district, Malawi: A retrospective record review

Dumisani Mfipa[1]*, Precious L. Hajison[2,3], Felistas Mpachika-Mfipa[4,5]

1 Agency for Scientific Research and Training, Lilongwe, Malawi, 2 Preluha Consultancy, Zomba, Malawi, 3 Pediatric and Child Health Association, Blantyre, Malawi, 4 Department of Nursing, Phalombe District Health Office, Phalombe, Malawi, 5 Centre for Reproductive Health, Kamuzu University of Health Sciences, Chichiri, Blantyre, Malawi

⊙ These authors contributed equally to this work.

* dumisanimfipa79@gmail.com

## Abstract

### Background

Birthweight has an impact on newborn's future health outcomes. Maternal factors, including age, delivery mode, HIV status, gestational age, parity and obstetric complications (pre-eclampsia or eclampsia [PE], antepartum hemorrhage [APH] and sepsis), however, have been shown as risk factors of low birthweight (LBW) elsewhere. For data-guided interventions, we aimed to identify predictors of LBW and compare newborn birthweights between different groups of maternal factors at Rev. John Chilembwe Hospital in Phalombe district, Malawi.

### Methods

Using a retrospective record review study design, we extracted data from maternity registers of 1244 women and their newborns from October, 2022 to March, 2023. Data were skewed. Median test was used to compare median birthweights. Chi-square or Fisher's exact tests were used to compare proportions of LBW among different groups of maternal factors. Multi-variable logistic regression with stepwise, forward likelihood method was performed to identify predictors of LBW.

### Results

Median birthweight was 2900.00g (interquartile range [IQR]: 2600.00g to 3200.00g). Prevalence of LBW was 16.7% (n = 208). Proportions of LBW infants were higher in women with PE, APH, including women with sepsis than controls (10 [47.6%] of 21 vs 7 [58.3%] of 12 vs 191 [15.8%] of 1211, $p < .001$). Lower in term and postterm than preterm (46 [5.5%] of 835 vs 2 [3.7%] of 54 vs 160 [45.1%] of 355, $p < .001$). The odds of LBW infants were higher in

**Data Availability Statement:** All relevant data are within the manuscript and its Supporting Information files.

**Funding:** The author(s) received no specific funding for this work.

**Competing interests:** The authors have declared that no competing interests exist.

preterm than term (AOR = 13.76, 95%CI: 9.54 to 19.84, $p$ < .001), women with PE (AOR = 3.88, 95%CI: 1.35 to 11.18, $p$ = .012), APH, including women with sepsis (AOR = 6.25, 95% CI: 1.50 to 26.11, $p$ = .012) than controls.

## Conclusion

Prevalence of LBW was high. Its predictors were prematurity, PE, APH and sepsis. Interventions aimed to prevent these risk factors should be prioritized to improve birthweight outcomes.

## Introduction

Birthweight has an important effect on the newborn's future health outcomes [1]. Low birthweight (LBW) is the main cause of morbidity and mortality among children aged below 1 year worldwide [2]. Prevalence of LBW, however, remains high in resource-constrained countries. World Health Organization (WHO) defines LBW as weight at birth of less than 2,500g, regardless of gestational age and sex [3]. It occurs due to either delivery of preterm or a growth restricted fetus [4]. Of particular importance, LBW is a vital indicator of maternal and child health service performance in a country [5]. Particularly, LBW is the main cause of morbidity and mortality among children aged below 5 years across the world [2,6]. It is the leading cause of perinatal mortality, as well as infant and childhood morbidity in the near future or beyond [4]. In sub-Saharan Africa (SSA), LBW is a strong determinant of mortality among newborn infants [7]. There are numerous long-term outcomes associated with both preterm and fetal growth restriction, mainly neurologic outcomes that include cerebral palsy, blindness, deafness and hydrocephaly [4]. Notably, information of neonatal deaths in South Africa included complications of LBW as well as prematurity and in the United Kingdom, neonatal deaths were more likely to occur due to complications of LBW and prematurity [5].

World Health Assembly in 2012, as a decision-making body of WHO, resolved that LBW prevalence be reduced by a target of at least 30% between 2012 and 2025 as part of the Comprehensive Implementation Plan on Maternal, Infant and Young Child Nutrition [8]. Evidence from studies done elsewhere suggest that birthweights of newborns vary according to maternal age, gestational age, HIV status, delivery mode, obstetric complications [antepartum hemorrhage, preeclampsia and sepsis] and parity [9–13]. In addition, a Malawian study reported that birthweights of newborns differ according to maternal HIV status [14].

Malawi has a prevalence of LBW estimated at 13.9% according to Malawi Multiple Indicator Cluster Survey (MICS) for 2019–20 Survey Findings Report [15]. This prevalence, however, varies depending on region, education, age at most recent live birth, place of delivery, birth order of most recent live birth, functional difficulties, ethnicity, wealth index quintile and area of residence, with rural areas having a prevalence of LBW estimated at 14.2%. Rev. John Chilembwe Hospital is located in a rural district of Phalombe in the southern region of Malawi. This hospital was officially opened in October, 2022. Unpublished data shows that the facility had registered 12% LBW deliveries from 10th October to 31st December 2022 and 9% LBW deliveries from 01st January to 31st March, 2023. In an effort to lower the prevalence of LBW, we conducted a study to compare newborn birthweights between different groups of maternal factors [age, delivery mode, HIV status, parity, gestational age and obstetric complications (preeclampsia or eclampsia, antepartum hemorrhage and sepsis)] and identified predictors of LBW among newborns of women delivering at this hospital. This locally-generated empirical

evidence will play a critical role in guiding medical experts at the hospital to design and implement evidence-backed and data-driven interventions. This will ensure delivery of quality maternal and child health services at this new facility.

We sought to explore the research question 'Do newborn birthweights differ significantly among different categories of maternal age, delivery mode, HIV status, gestational age of pregnancy, parity and obstetric complications (preeclampsia or eclampsia, antepartum hemorrhage and sepsis) at Rev. John Chilembwe Hospital in Phalombe district, Malawi?' This study established statistically significant differences in median birthweights of newborns between different maternal ages, parity, obstetric complications and gestational age of pregnancy. We also explored the research question 'What are the predictors of LBW among newborns delivered at Rev. John Chilembwe Hospital in Phalombe district, Malawi?' We observed that prematurity, preeclampsia and eclampsia, antepartum hemorrhage and sepsis were statistically associated with LBW.

## Materials and methods

### Study design and setting

We used a retrospective record review study design. The study was conducted at Rev. John Chilembwe Hospital in Phalombe district in southern Malawi.

### Study participants

The study participants were newborns and their respective mothers. The authors had access to maternity registers which had information that could identify individual participants during data collection. However, all data were fully anonymized after data collection such that no single person shall be able to identify individual participants.

**Sample and procedures.**   All live births from 10th October, 2022 (the first day the facility started offering maternity services) to 31st March, 2023 at Rev. John Chilembwe Hospital in Phalombe district, including those referred from other health facilities were included in the study. Data were accessed between 01st and 10th August, 2023 for research purposes. Data related to newborn birthweights, maternal age, delivery mode, HIV status, gestational age, parity and obstetric complications (preeclampsia or eclampsia, antepartum hemorrhage and sepsis) were extracted from the maternity registers.

**Inclusion and exclusion criteria.**   All live births resulting from singleton pregnancies were included in the study except those newborns with missing birthweights. United Nations Children's Fund (UNICEF) and WHO define incidence of LBW in a population as the percentage of live births that weigh less than 2,500g out of the total of live births during the same time period [16]. We, therefore, excluded all fresh stillbirths and macerated stillbirths. In addition, mothers of the sampled newborns with incomplete data were excluded from the study.

### Ethics approval and consent to participate

The research was performed in accordance with the Declaration of Helsinki and written ethical approvals were obtained from the Phalombe District Health Office (DHO) Research Committee and the Malawi National Health Sciences Research Committee (NHRSC) [Approval Number 4131]. This was a retrospective study and both Committees waived the requirement for informed consent.

## Data collection

We developed a data extraction checklist that reflected the components of maternity registers and used it as a guide to extract data from the registers.

## Study variables and measurements

The outcome variable was birthweight of newborns, which was measured as a continuous variable but categorized and coded as LBW: <2,500g (1) or non-LBW: ≥2,500g (0) whereas independent variables were maternal age which was categorized and coded as ≤19 years (1 = reference category), 20–24 years (2), 25–34 years (3) and ≥35 years (4)]; Delivery mode was categorized and coded as spontaneous vertex delivery (SVD) and breech delivery (normal birth) [delivery of the baby through the birth canal without any surgery and use of instruments] (1 = reference category), assisted vaginal delivery (AVD) [vacuum extraction delivery] (2) and caesarean section (CS) delivery [delivery of a newborn through surgical incisions made through the abdominal wall] (3); HIV status: HIV- [women without HIV as confirmed by a new or previous HIV test] (1 = reference category) and HIV+ [women with HIV as confirmed by a new or previous HIV test] (2); parity: primiparous women [women giving birth for the first time] (1 = reference category), multiparous women [women giving birth for second time, third time or fourth time] (2) and grandmultiparous women [women giving birth for the fifth time or more] (3) and obstetric complications: women with preeclampsia, including those with eclampsia (1), women with antepartum hemorrhage (APH), including those with sepsis (2), control group [women with no preeclampsia or eclampsia, antepartum hemorrhage and sepsis] (3 = reference category) and gestational age of pregnancy was categorized and coded as: term [newborns delivered between 37 weeks, 0 days and 41 weeks, 6 days] (1 = reference category), preterm [newborns delivered at or before 36 weeks, 6 days] (2) and postterm [newborns delivered at or after 42 weeks, 0 days] (3).

## Statistical analysis

Data were entered and analyzed using IBM SPSS Statistics for Windows Version 20.0 (IBM Corp., Armonk, NY, USA). Chi-square or Fisher's exact tests were used to compare the proportions of LBW among different groups of maternal factors. Normality tests by Shapiro-Wilk and Kolmogorov-Smirnov tests were conducted. We concluded that data did not follow a normal distribution. We, therefore, used median test to compare median birthweights of newborns as it is robust to outliers present in our data. For multiple pairwise comparisons, we used median test post hoc test called the "Pairwise median test" to test differences in median birthweight of newborns among different groups of maternal factors. We reported $p$ values that were adjusted using the Bonferroni correction. The $p$ value of less than .05 was considered as the statistical significance for all analyses. We reported median birthweights of newborns with their corresponding interquartile ranges (IQR). Median test is a special case of Pearson's Chi-squared test [17]. Thus, for 2 × 2 contingency tables, we estimated the effect sizes using phi ($\varphi$) [18]:

$$\varphi = \sqrt{\frac{\chi^2}{n}}$$

Where, $\chi^2$ = Chi-square statistic and $n$ = the number of observations.

For larger contingency tables (non $2 \times 2$ contingency tables), we estimated effect sizes using Cramér's $V$ [18]:

$$V = \sqrt{\frac{\chi^2}{n \cdot df^*}}$$

Where, $df^* = \min{(r - 1, c - 1)}$ and $r$ = the number of rows and $c$ = the number of columns in the contingency table. In other words, $df^* = k—1$ and $k$ is the number of rows or columns in the table, whichever is smaller.

First, we converted Cramér's $V$ to Cohen's omega ($\omega$) by multiplying Cramér's $V$ by the square root of the degrees of freedom of the contingency table (the length of the minimum dimension) [18];

$$\omega = V\sqrt{df*}$$

We, then, interpreted the estimated effect sizes according to Cohen's omega ($\omega$) benchmarks as follows: small ($\omega = 0.1$), medium ($\omega = 0.3$) and large ($\omega = 0.5$) effects [19] as $df^*$ was equivalent to 1 in our study. Multivariable logistic regression with stepwise, forward likelihood selection analyses was performed to identify predictors of LBW after adjusting for all variables which were significant at univariable level ($p < .05$). Hosmer-Lemeshow goodness of fit test was used to check for a logistic model fit. The $p$ value of $>.05$ indicated a good fitting model as it failed to reject the null hypothesis (null hypothesis: there is no difference between the observed and model-predicted values). Hence, this indicated that the model-predicted values fitted the observed data at an acceptable level. The results of univariable and multivariable analyses were reported as crude odds ratios (CORs) and adjusted odds ratios (AORs) with 95% confidence intervals (CIs), respectively.

## Results

This study had 1,244 women (of which 34.3% were adolescent girls) who met the inclusion criteria. Eighty-seven (87) women did not meet the inclusion criteria as 69 had their gestation weeks either unrecorded or unclear, 13 had their parity unclear, 4 had their ages unrecorded and 1 had her HIV status unrecorded. One hundred eighty-eight (188) newborns were not included as 47 had birthweights unrecorded, 104 were twins and 37 were stillbirths (macerated stillbirths = 14 and fresh stillbirths = 23).

### Comparison of proportions of LBW among different groups of maternal factors at Rev. John Chilembwe Hospital in Phalombe district, Malawi

Table 1 shows that the proportion of LBW was higher in primiparous women than multiparous and grandmultiparous women (19.9% vs 15.1% vs 11.6%, $p = .014$) and lower in term and postterm than preterm newborns (5.5% vs 3.7% vs 45.1%, $p < .001$). The results further showed that it was higher in women with preeclampsia, including those with eclampsia and women with antepartum hemorrhage, including those with sepsis than the control group (47.6% vs 58.3% vs 15.8%, $p < .001$). We further observed that the proportion of LBW deliveries was higher in adolescent girls than in young women, older women and women of advanced maternal age (20.4% vs 16.4% vs 15.4% vs 10.4%, $p = .026$). No significant differences in LBW deliveries were identified among women who delivered through normal birth, assisted birth and caesarean section birth (17.4% vs 14.3% vs 14.3%, $p = .504$) as well as between newborns of HIV- and HIV+ women (16.7% vs 16.8%, $p = .979$).

**Table 1. Comparison of proportions of LBW among different groups of maternal factors at Rev. John Chilembwe Hospital in Phalombe district, Malawi.**

| Maternal factor | LBW | Non–LBW | Total (n = 1,244) | Chi-square | df | P value |
|---|---|---|---|---|---|---|
| Age group (years) | | | | 9.28 | 3 | **.026** |
| ≤19 | 87 (20.4%) | 340 (79.6%) | 427 (34.3%) | | | |
| 20–24 | 57 (16.4%) | 290 (83.6%) | 347 (27.9%) | | | |
| 25–34 | 47 (15.4%) | 259 (84.6%) | 306 (24.6%) | | | |
| ≥35 | 17 (10.4%) | 147 (89.6%) | 164 (13.2%) | | | |
| Delivery mode | | | | 1.37 | 2 | .504 |
| Normal birth | 170 (17.4%) | 809 (82.6%) | 979 (78.7%) | | | |
| Assisted birth | 1 (14.3%) | 6 (85.7%) | 7 (0.6%) | | | |
| Caesarean section birth | 37 (14.3%) | 221 (85.7%) | 258 (20.7%) | | | |
| Obstetric complications‡‡ | | | | | | < .001† |
| Preeclampsia/eclampsia | 10 (47.6%) | 11 (52.4%) | 21 (1.7%) | | | |
| Antepartum hemorrhage/sepsis | 7 (58.3%) | 5 (41.7%) | 12 (1.0%) | | | |
| Control group‡ | 191 (15.8%) | 1,020 (84.2%) | 1,211 (97.3%) | | | |
| HIV status | | | | 0.001 | 1 | .979 |
| HIV- | 188 (16.7%) | 937 (83.3%) | 1,125 (90.4%) | | | |
| HIV+ | 20 (16.8%) | 99 (83.2%) | 119 (9.6%) | | | |
| Parity | | | | 8.48 | 1 | **.014** |
| Primiparous | 111 (19.9%) | 447 (80.1%) | 558 (44.9%) | | | |
| Multiparous | 75 (15.1%) | 422 (84.9%) | 497 (40.0%) | | | |
| Grandmultiparous | 22 (11.6%) | 167 (88.4%) | 189 (15.1%) | | | |
| Gestational age of pregnancy | | | | 286.85 | 2 | **< .001** |
| Term | 46 (5.5%) | 789 (94.5%) | 835 (67.1%) | | | |
| Preterm | 160 (45.1%) | 195 (54.9%) | 355 (28.6%) | | | |
| Postterm | 2 (3.7%) | 52 (96.3%) | 54 (4.3%) | | | |
| **Total** | **208 (16.7%)** | **1,036 (83.3%)** | **1,244 (100.0%)** | | | |

Bold values denote statistical significance at the *p* value < .05 level.

†Fisher's exact test was used for analysis as appropriate.

‡‡Obstetric complications in this study refer to preeclampsia or eclampsia, antepartum hemorrhage and sepsis.

‡Control group in this study refers to women without preeclampsia or eclampsia, antepartum hemorrhage and sepsis.

LBW stands for low birthweight.

### Normality tests using Kolmogorov-Smirnov and Shapiro-Wilk tests

Table 2 shows the results of the Kolmogorov-Smirnov and Shapiro-Wilk tests to ascertain the distribution of the data. Normality tests conducted by Kolmogorov-Smirnov and Shapiro-Wilk tests were significant for all categories of maternal factors with the exception of the following categories; 35 years and above, assisted birth, preeclampsia and eclampsia, antepartum hemorrhage and sepsis as well as postterm. Normality tests conducted by Kolmogorov-Smirnov and Shapiro-Wilk tests were significant for the outcome variable of newborn birthweights. We, therefore, concluded that data in our study did not follow a normal distribution.

### Median birthweight of newborns at Rev. John Chilembwe Hospital in Phalombe district, Malawi

Our results show that the median birthweight of newborns was 2900.00g (interquartile range [IQR]: 2600.00g to 3200.00g) between October, 2022 and March, 2023 at Rev. John Chilembwe Hospital in Phalombe district, Malawi.

**Table 2. Normality tests using Kolmogorov-Smirnov and Shapiro-Wilk tests.**

| Independent Variable | Normality test/Kolmogorov-Smirnov test | *P* value | Normality test/Shapiro-Wilk test | *P* value |
|---|---|---|---|---|
| Maternal age (years) | | | | |
| ≤19 | 0.070 | < .001 | 0.987 | **.001** |
| 20–24 | 0.097 | < .001 | 0.965 | < .001 |
| 25–34 | 0.070 | **.001** | 0.985 | **.002** |
| ≥35 | 0.065 | .087 | 0.986 | .093 |
| Delivery mode | | | | |
| Normal birth | 0.072 | < .001 | 0.984 | < .001 |
| Assisted birth | 0.203 | .200 | 0.884 | .245 |
| Caesarean section birth | 0.098 | < .001 | 0.971 | < .001 |
| Obstetric complications‡‡ | | | | |
| Preeclampsia/eclampsia | 0.137 | .200 | 0.971 | .758 |
| Antepartum hemorrhage/sepsis | 0.173 | .200 | 0.951 | .654 |
| Control group‡ | 0.069 | < .001 | 0.984 | < .001 |
| HIV status | | | | |
| HIV- | 0.068 | < .001 | 0.984 | < .001 |
| HIV+ | 0.108 | **.002** | 0.973 | **.015** |
| Parity | | | | |
| Primiparous | 0.075 | < .001 | 0.980 | < .001 |
| Multiparous | 0.066 | < .001 | 0.988 | **.001** |
| Grandmultiparous | 0.131 | < .001 | 0.960 | < .001 |
| Gestational age of pregnancy | | | | |
| Term | 0.072 | < .001 | 0.981 | < .001 |
| Preterm | 0.078 | < .001 | 0.979 | < .001 |
| Postterm | 0.101 | .200 | 0.964 | .100 |
| **Outcome Variable** | | | | |
| Newborn birthweight (in grams) | 0.072 | < .001 | 0.983 | < .001 |

Bold values denote statistical significance at the *p* < .05 level.

‡‡Obstetric complications in this study refer to preeclampsia or eclampsia, antepartum hemorrhage and sepsis.

‡Control group in this study refer to women without preeclampsia or eclampsia, antepartum hemorrhage and sepsis.

## Comparison of newborn birthweights among different groups of maternal factors at Rev. John Chilembwe Hospital in Phalombe district, Malawi

Table 3 shows the results of the median test. There was a statistically significant difference in median birthweights among newborns of adolescent girls (≤19 years), young women (20–24 years), older women (25–34 years) and women of advanced maternal age (≥35 years), ($\chi^2$ [3] = 18.08, *p* < .001, ω = 0.12). Data also showed a statistically significant difference in median birthweights between newborns of women with preeclampsia, including those with eclampsia, women with antepartum hemorrhage, including those with sepsis and the control group, ($\chi^2$ [2] = 13.32, *p* = .001, ω = 0.10). Our results further reported a statistically significant difference in median birthweights among newborns of primiparous, multiparous and grandmultiparous women, ($\chi^2$ [2] = 18.95, *p* < .001, ω = 0.12) as well as among term, preterm and postterm newborns, ($\chi^2$ [2] = 226.66, *p* < .001, ω = 0.43). No statistically significant differences were observed between infants of HIV- and HIV+ women, ($\chi^2$ [1] = 1.446, *p* = .229, ω = 0.03). We observed no statistically significant difference in median birthweights among newborns delivered through normal, assisted vaginal and caesarean section births ($\chi^2$ [2] = 5.11, *p* = .076, ω = 0.06).

**Table 3. Comparison of newborn birthweights among different groups of maternal factors at Rev. John Chilembwe Hospital in Phalombe district, Malawi.**

| Maternal factor | Median birthweight | Interquartile range (IQR) | Chi-square | df | P-value | *Cohen's ω |
|---|---|---|---|---|---|---|
| Age group (years) | | | 18.08 | 3 | **< .001** | 0.12 |
| ≤19 | 2800.00g | 2500.00g to 3100.00g | | | | |
| 20–24 | 2900.00g | 2600.00g to 3200.00g | | | | |
| 25–34 | 2942.50g | 2600.00g to 3300.00g | | | | |
| ≥35 | 3000.00g | 2700.00g to 3400.00g | | | | |
| Delivery mode | | | 5.11 | 2 | .076[✦] | 0.06 |
| Normal birth | 2900.00g | 2600.00g to 3200.00g | | | | |
| Assisted birth | 3200.00g | 2800.00g to 3500.00g | | | | |
| Caesarean section birth | 3000.00g | 2700.00g to 3300.00g | | | | |
| Obstetric complications‡‡ | | | 13.32 | 2 | **.001** | 0.10 |
| Preeclampsia/eclampsia | 2500.00g | 2100.00g to 2850.00g | | | | |
| Antepartum hemorrhage/sepsis | 2137.50g | 1825.00g to 2725.00g | | | | |
| Control group‡ | 2900.00g | 2600.00g to 3200.00g | | | | |
| HIV status | | | 1.446 | 1 | .229 | 0.03 |
| HIV- | 2900.00g | 2600.00g to 3200.00g | | | | |
| HIV+ | 2900.00g | 2600.00g to 3200.00g | | | | |
| Parity | | | 18.95 | 2 | **< .001** | 0.12 |
| Primiparous | 2800.00g | 2500.00g to 3100.00g | | | | |
| Multiparous | 2900.00g | 2600.00g to 3300.00g | | | | |
| Grandmultiparous | 3000.00g | 2700.00g to 3300.00g | | | | |
| Gestational age of pregnancy | | | 226.66 | 2 | **< .001** | 0.43 |
| Term | 3000.00g | 2700.00g to 3300.00g | | | | |
| Preterm | 2500.00g | 2200.00g to 2800.00g | | | | |
| Postterm | 3400.00g | 3100.00g to 3700.00g | | | | |

Bold values denote statistical significance at the *p* value < .05 level.

*Cohen's, ω measures the effect size.

[✦]An exact two-sided *p* value of Pearson chi-square test was used as appropriate.

‡‡Obstetric complications in this study refer to preeclampsia or eclampsia, antepartum hemorrhage and sepsis.

‡Control group in this study refers to women without preeclampsia or eclampsia, antepartum hemorrhage and sepsis.

**Pairwise comparison of the median birthweights of the newborns.** Pairwise comparison showed a statistically significant difference in median birthweights between newborns of adolescent girls and older women (*p* = .013, adjusted using the Bonferroni correction), adolescent girls and women of advanced maternal age (*p* = .001, adjusted using the Bonferroni correction). We observed no statistically significant differences in median birthweights of newborns between adolescent girls and young women (*p* = .634, adjusted using the Bonferroni correction) as well as young women and older women (*p* > .99, adjusted using the Bonferroni correction) as well as young women and women of advanced maternal age (*p* = .282, adjusted using the Bonferroni correction). Data showed no statistically significant differences in median birthweights of newborns between older women and women of advanced maternal age (*p* > .99, adjusted using the Bonferroni correction).

Our results showed a statistically significant difference in median birthweights between term and preterm newborns (*p* < .001, adjusted using the Bonferroni correction), term and postterm newborns (*p* < .001, adjusted using the Bonferroni correction) as well as preterm and postterm newborns (*p* < .001, adjusted using the Bonferroni correction).

We observed a significant difference in median birthweights of the newborns between women with preeclampsia, including those with eclampsia and the control group ($p = .034$, adjusted using the Bonferroni correction) as well as antepartum hemorrhage, including those with sepsis and the control group ($p = .023$, adjusted using the Bonferroni correction). Data showed no statistically significant differences in median birthweights between newborns of women with preeclampsia, including those with eclampsia and women with antepartum hemorrhage, including those with sepsis ($p > .99$, adjusted using the Bonferroni correction).

We also found a statistically significant difference in median birthweights between newborns of primiparous and multiparous women ($p = .002$, adjusted using the Bonferroni correction) as well as newborns of primiparous and grandmultiparous women ($p = .013$, adjusted using the Bonferroni correction). No statistically significant difference in median birthweights was observed between newborns of multiparous and grandmultiparous women ($p > .99$, adjusted using the Bonferroni correction).

## Predictors of low birthweight at Rev. John Chilembwe Hospital in Phalombe district, Malawi

Table 4 shows that the odds of LBW were 13.76 times higher in preterm than term newborns (adjusted OR = 13.76, 95%CI: 9.54 to 19.84, $p < .001$). Our data also suggested that the odds of LBW were 3.88 times higher in newborns of women with preeclampsia, including those with eclampsia than the control group (adjusted OR = 3.88, 95%CI: 1.35 to 11.18, $p = .012$). Data

**Table 4. Predictors of low birthweight at Rev. John Chilembwe Hospital in Phalombe district, Malawi.**

| Maternal factors | Univariable | | | Multivariable† | | |
|---|---|---|---|---|---|---|
| | Odds ratio | 95% Confidence interval | *P* value | Odds ratio | 95% Confidence interval | *P* value |
| Maternal age (years) | | | | | | |
| ≤19 | Reference | | | | | |
| 20–24 | 0.77 | 0.53 to 1.11 | .161 | - | - | - |
| 25–34 | 0.71 | 0.48 to 1.05 | .084 | - | - | - |
| ≥35 | 0.45 | 0.26 to 0.79 | **.005** | - | - | - |
| Obstetric complications‡‡ | | | | | | |
| Preeclampsia/eclampsia | 4.86 | 2.03 to 11.59 | **< .001** | 3.88 | 1.35 to 11.18 | **.012** |
| Antepartum hemorrhage/sepsis | 7.48 | 2.35 to 23.80 | **.001** | 6.25 | 1.50 to 26.11 | **.012** |
| Control group‡ | Reference | | | Reference | | |
| Parity | | | | | | |
| Primiparous | Reference | | | | | |
| Multiparous | 0.72 | 0.52 to 0.99 | **.042** | - | - | - |
| Grandmultiparous | 0.53 | 0.33 to 0.87 | **.011** | - | - | - |
| Gestational age of pregnancy | | | | | | |
| Term | Reference | | | Reference | | |
| Preterm | 14.07 | 9.79 to 20.24 | **< .001** | 13.76 | 9.54 to 19.84 | **< .001** |
| Postterm | 0.66 | 0.16 to 2.79 | .572 | 0.64 | 0.15 to 2.74 | .547 |

Bold values denote statistical significance at the *p* value < .05 level.

‡‡Obstetric complications in this study refer to preeclampsia or eclampsia, antepartum hemorrhage and sepsis.

‡Control group in this study refer to women without preeclampsia or eclampsia, antepartum hemorrhage and sepsis.

†adjusted for maternal age, obstetric complications, parity and gestational age in the multivariable model.

NOTE: delivery mode and HIV status were not significant at univariable level.

Hosmer and Lemeshow Goodness of fit for final model: Chi-square = 0.397, p-value = .820.

Pseudo R-squared = 0.332.

further suggested that the odds of LBW were 6.25 times higher in newborns of women with antepartum hemorrhage, including those with sepsis than the control group (adjusted OR = 6.25, 95%CI: 1.50 to 26.11, *p* = .012).

## Discussion

This study was aimed at comparing birthweights of newborns between different groups of maternal factors [age, HIV status, parity, delivery mode, gestational age and obstetric complications (preeclampsia or eclampsia, antepartum hemorrhage and sepsis). It was also aimed at identifying predictors of LBW of newborns delivered at Rev. John Chilembwe Hospital in Phalombe district in Malawi.

Our study reported a statistically significant difference in newborn birthweights among different maternal age groups. Specifically, we observed significant differences in median birthweights of newborns of adolescent girls (≤19 years) and older women (25–34 years) as well as those of adolescent girls and women of advanced maternal age (≥35 years). It is important to note that these differences were not clinically significant. We further observed that the higher the maternal age group, the larger the median birthweights of the newborns, and vice-versa. Our study found that the odds of LBW were lower in women of advanced maternal age than the adolescent girls in the univariable analysis. We, however, observed that maternal age was not significantly associated with LBW in the multivariable analysis. This is consistent with findings from a study on determinants of LBW among deliveries at a Referral Hospital in Northern Ethiopia which also observed that maternal age was not significantly associated with LBW [20]. Research findings have been inconsistent on the effects of maternal age on birthweight. This inconsistency can be attributed to other factors. For example, maternal age may also be related to racial and socio-economic differences which may confound research findings and distort conclusions about the effect of age on birthweight [21,22]. Our study did not adjust for socio-economic differences of the study participants as we analyzed secondary data which did not capture this variable. Several studies, however, have shown that maternal age is a predictor of LBW [7,9,23,24]. However, there are inconsistences regarding its effects on birthweight. For instance, an Ethiopian study suggested that maternal age was a protective factor for LBW as an increase in age by one year resulted in a 4% risk reduction (AOR = 0.96, 95%CI: 0.92, 1.00) [24]. Likewise, an observational sub Saharan Africa multi-country study reported that young maternal age increases the risk for adverse pregnancy outcomes and that it was a strong determinant of LBW among newborn infants [7]. An Iranian study, however, observed that maternal age at delivery of less than 18 years and above 35 years was associated with an increased risk of LBW [23]. A study on the effects of maternal age and parity on birthweight of newborns of mothers with term and singletons in Ethiopia also observed that women with a maternal age of 40 and above were associated with a higher risk of delivering LBW newborns compared to a maternal age of 30–34 [9]. The study reported the following mean birthweights of newborns; 3,159.1g (SD = 518.8), *p* = .765 (19 years and below) vs 3,144.1g (SD = 514.8), *p* = .718 (20–24 years) vs 3,168.1g (SD = 506.2), *p* = .299 (25–29 years) vs 3,004.2g (SD = 589.9), *p* < .001 (30–34 years) vs 2,844.7g (SD = 664.3), *p* = .052 (35–39 years) vs 2,397.8g (SD = 673.5), *p* = .006 (40 years and above) [9].

About thirty-four percent (427 [34.3%] of 1244) women in our study were adolescent girls. This finding is concerning as several studies have associated young maternal age with adverse pregnancy outcomes. Our finding is consistent with previous findings which showed that fertility rate for adolescents in Malawi is one of the highest in the SSA region at 143 births per 1,000 girls aged 15–19 years [25] with 29% of adolescent girls (15–19 years) having begun child-bearing [26]. Twenty-six percent of all pregnancies reported through the formal health

system are among 15–19 years in Malawi [27]. Malawi has one of the earliest age of first sex in the world where most young people start having consensual sex at the age of 15, on average [26]. An early sexual debut among girls is more common especially in rural areas located in the southern region of Malawi [28], where this study was conducted. Our findings showed that Rev. John Chilembwe Hospital in Phalombe district recorded a high prevalence of LBW estimated at 16.7%. The proportion of LBW babies was higher in adolescent girls than in other individual maternal age groups. The higher proportion of LBW babies in adolescents (87 [20.4%] of 427) in this study can be attributed to biological immaturity as the adolescent is still growing and developing and may compete for nutrient with the unborn baby [29]. There is an important need for concerted efforts among stakeholders and partners to end child marriages, adolescent pregnancies and child births among adolescent girls in the district. Hence, we further recommend that adolescent SRHR services should be scaled up to increase access to and usage of contraceptives to deter adolescent pregnancies and child bearing among adolescent girls which contribute significantly to low birthweights at Rev. John Chilembwe Hospital. We observed a higher prevalence of LBW than the one reported for the study period by the hospital. This difference came about due to data cleaning challenges among midwives. In few instances, midwives did not indicate that babies were LBW even though they had entered the newborn birthweights as less than 2,500g. This led to incorrect LBW summations which resulted in lower prevalence of LBW. With reference to WHO definition of LBW, we rectified such errors by correctly categorizing LBW as those newborns weighing less than 2,500g since the data on birthweight was available in the maternity registers. Poor data quality on LBW was also noted in the Malawi Multiple Indicator Cluster Survey (MICS) 2019–2020 Survey Findings Report, and it recommended cautious use of this data as it was considered an underestimation of the true prevalence [15].

We also observed that there was no statistically significant difference in median birthweight of newborns between the different maternal modes of delivery. This finding is in contrast with an Iranian study which observed a statistically significantly lower mean birthweight of newborns in caesarean section delivery group of 3, 166g (± 442.4) as compared to 3, 213g (± 454.8) in normal vaginal delivery group ($p < .001$) [12]. However, the differences were not clinically significant. A caesarean section delivery maybe necessary if there are concerns for mother's or baby's safety. Data in our study suggests that compared to normal vaginal births, assisted vaginal births and caesarean sections were modes of delivery for big babies. This is in agreement with a systematic review and meta-analysis study which reported that big babies had high odds of being delivered through an emergency caesarean section [30]. Furthermore, obstructed or prolonged labor is a common indication for caesarean section delivery [31]. Studies, however, indicate that the risk of caesarean section delivery escalates with an increase in birthweight but the proportion of assisted vaginal births decreases with increasing birthweights [32,33]. About 0.6% of deliveries were assisted vaginal births at Rev. John Chilembwe Hospital in Phalombe district in Malawi. WHO recommends that caesarean sections should not exceed 15% of all deliveries [34]. Caesarean sections accounted for 20.7% of all deliveries in the current study. This discrepancy could be attributed to referrals due to unavailability of Comprehensive Emergency Obstetric and Newborn Care (CEmONC) services at health centers. WHO, further, recommends that caesarean sections should only be provided for medical reasons and all deserving women should have access to caesarean section without necessarily making obstetricians striving to achieve a specific rate [34].

Our results showed that newborns of women with preeclampsia, including those with eclampsia, women with antepartum hemorrhage, including those with sepsis had a significantly lower median birthweight than those of women without these kinds of complications. This is similar to a prospective cohort study in a tertiary referral center in urban Uganda that

observed a lower mean birthweight of 2.48 kg (SD = 0.81) for preeclampsia cases than 3.06 kg (SD = 0.46) for controls ($p <$ .001) [13]. We encourage health facilities to prioritize as well as intensify comprehensive physical examination and screening for obstetric complications, including preeclampsia or eclampsia, antepartum hemorrhage and sepsis among pregnant women attending ANC services to prevent development of these complications and identify the high-risk cases and offer appropriate management. In this regard, lessons can be drawn from a study on utilization of cervical cancer screening (CCS) services which observed that the proportion of women who had done CCS was significantly higher among those who were recommended by health workers than those who were not [35]. Patient-centered care that focusses on pregnancy-related complications like preeclampsia, antepartum hemorrhage and sepsis may afford many pregnant women an opportunity to receive quality care before delivery and improve birthweights of newborns in the district.

We, further, observed that women with preeclampsia, including eclampsia and women with antepartum hemorrhage, including sepsis were more likely to deliver newborns with LBW as compared to the control group. This is in agreement with a Qatari study that observed that the distribution of antepartum hemorrhage was significantly different between mothers of LBW and mothers of normal birth weight (NBW) ($\geq$2500 g) babies (21.4% versus 14.5%, $p$ = .046) [36]. That study further reported that mothers with antepartum hemorrhage were significantly associated with delivering newborn babies with LBW after adjusting for gestational age (AOR = 1.6, 95%CI: 1.1–2.5, $p$ = .048) [36].

Studies have shown that maternal sepsis is associated with delivering of newborns with LBW. For instance, a systematic review and meta-analysis study on the relationship between Pregnancy-Associated Malaria (PAM) and adverse pregnancy outcomes reported that the overall risk of LBW was 63% higher among women with PAM than women without PAM (95%CI: 1.48–1.80) [37]. Urinary tract infections (UTIs) have also been associated with adverse pregnancy outcomes such as increased rates of LBW [38]. This is in agreement with findings from a Pakistani study that reported that UTI has a significant impact on pregnancy outcome, especially premature labor and LBW [39]. In addition to that, another study observed that maternal urinary tract infection presented 4.24-fold higher chance of late-onset sepsis in very LBW (VLBW) [40]. Our study, therefore, suggest that emphasis should be on the delivery of quality maternal care services such as screening and prompt treatment of obstetric complications.

We further observed that there were no statistically significant differences in median birthweights of newborns born to HIV positive and HIV negative women. Similar findings were reported in a Tanzanian study which reported that treated HIV+ women had a risk similar to that of the HIV- women for all outcomes, including birthweight, prematurity, small-for-gestational-age (SGA), and perinatal mortality except for low Apgar score [41]. A systematic review and meta-analysis on perinatal outcomes associated with maternal HIV infection observed that no association was identified between maternal HIV infection and very preterm birth, very SGA, very LBW and miscarriage [42]. This is in contrast with findings from a previous Malawian study that analyzed DHS data for 2010 which found that the mean birthweight of infants born to HIV positive mothers was statistically significantly lower compared to infants born to HIV negative mothers (3,192g [SD = 724] vs 3,275g [SD = 723], $p$ =. 0412) [14]. Likewise, the frequency of LBW infants among HIV negative mothers was statistically significantly lower from their HIV positive counterparts (9.26% vs 13.6%, $p$ = .0337) [14]. Further, it was observed that HIV+ women who had not received antiretroviral therapy were associated with preterm birth, LBW, SGA and stillbirth, especially in SSA [42]. Our finding can be attributed to the fact that all HIV+ women in our study were on ART treatment due to a robust, well-coordinated and effective Prevention of Mother to Child Transmission (PMTCT) of HIV

Option B+ program, which has greatly improved health outcomes of newborns and mothers over the years in Malawi in general and in Phalombe district in particular. We, therefore, encourage health authorities to intensify PMTCT programs in the district, including in hard-to-reach areas to sustain the achievements made and maximize the benefits this program provides to HIV exposed newborns and HIV infected mothers. We, also, urge health authorities to continue sensitizing pregnant women on the importance of early antenatal care (ANC) as this offers an opportunity to ascertain the HIV status and linkage to care in good time hence the improved health status of the mother and prevention of possible adverse effects to the newborn, including low birthweight [43].

Our results showed that newborns born to primiparous mothers had a significantly lower median birthweight as compared to those born to multiparous and grandmultiparous women. However, this difference was not clinically significant. This finding is similar to a Polish study found that the mean birthweights of neonates born to primiparous mothers and multiparous mothers were 3,356.3g (SD = 524.9) and 3,422.7g (SD = 538.6), respectively ($p \leq .001$) [10]. Similarly, a different Polish study and an India study observed that primiparous mothers had a statistically significant higher frequency of LBW than multiparous mothers [44,45]. This is in contrast to the findings from an Ethiopian study which observed that primiparous women had less risk of having an LBW baby compared to multiparous women [9]. The difference in newborn birthweights between primiparous and multiparous mothers in the present study could be due to the physiological changes that occur in subsequent pregnancies as compared to primigravidas hence the known fact that primiparity increases the risk of LBW [46]. Biologically, among other factors, the uteroplacental blood flow is lesser in first pregnancy compared to subsequent pregnancies and structural factors which limit uterine capacity are present in first pregnancy [47]. We suggest that during antenatal care, women with first pregnancy should be offered all the interventions in the antenatal care package to prevent increasing the likelihood of a LBW baby.

Further to that, it was observed at univariate analysis that multiparous and grandmultiparous women were less likely to deliver LBW babies than primiparous women. This is consistent with results from studies elsewhere. This significance, however, disappeared at multivariable analysis level. Elsewhere, studies have also reported that primiparous women were likely to deliver newborns with LBW as compared to multiparous women [48]. It is, therefore, important to consider this group of women when designing programs aimed at preventing deliveries of newborns with LBW.

We observed that preterm newborns had a significantly lower median birthweight than term newborns whereas postterm newborns had a significantly higher median birthweight than term newborns. This is consistent with findings from studies elsewhere. For example, an Indian study observed a statistically significant lower mean birthweight in preterm babies than term babies ($1860 \pm 442.04$g vs $2570 \pm 400.72$g, [t = 18.43, $p < .001$]) whereas postterm babies had a significantly higher mean birthweight than term babies ($2785 \pm 300.09$g, [t = 6.44, $p < .001$]) [11]. Premature babies are born before completion of their term as a result of other factors, including medical and gynecological factors do not complete their normal physical development inside their mothers' womb and are more likely to be LBW [20]. Our study also observed a medium effect size in relation to the differences in median birthweights among newborns of different gestational ages at birth. It is important, therefore, that efforts should be made to design and implement policies and strategies aimed at preventing preterm deliveries in order to lower LBW. Furthermore, improvements in determining accurate gestational age in pregnancy should be made, and that ultrasound scanning should be used more to assess gestational age. This will ensure that almost all preterm deliveries are identified prior to delivery and that the management of a LBW baby is anticipated and planned for.

We also found that being a preterm baby was significantly associated with LBW. This is consistent with findings from several studies. For instance, an Ethiopian study observed that the odds of LBW were higher in preterm births than in term births (AOR = 5.32, 95%CI: 2.959–9.567) [20]. Similarly, a systematic review and meta-analysis of several studies on low birth weight and its associated factors in Ethiopia reported that short gestation (preterm birth) was significantly associated with LBW (AOR = 6.4, 95%CI: 2.5–10.3) [49]. This highlights the need for midwives and obstetricians to specifically target pregnant women with services that may address preterm deliveries in their health facilities.

## Limitations of the study

This was a retrospective record review study design as such we relied on data that was recorded in maternity registers with some variables having incomplete and missing data. This led to the exclusion of some mothers and their newborns from the study. Nonetheless, the large sample of the data that was collected and analyzed compensated for those that did not meet the inclusion criteria. Our selection of independent variables was also guided by and restricted to information that was available in the maternity registers. This meant that we could not analyze socio-demographic data, maternal weight as well as lifestyle data such as smoking, alcohol consumption and drug abuse as these registers do not capture such data. The results of this study should be generalized with caution as the study was conducted at one facility in the district. However, this is a referral hospital for the district such that it was highly likely that newborns of mothers referred from other facilities within the district were represented in the study. Our study included gestational age of pregnancy as an important maternal factor associated with LBW. Gestational age of pregnancy, however, is mostly calculated from the last menstrual period (LMP) and fundal height which in most cases is not reliable as some women do not recall the dates and some midwives do not correctly assess the fundal height. The use of ultra sound scanning in this hospital set up in determining gestational age is not consistent. The other better way of determining the gestational age is performing a neuromuscular assessments of the neonate using the Ballard score, however, this also is not routinely done and is not recorded in the maternity register. The maternity registers from where we extracted data were designed in such a way that a woman with either preeclampsia or eclampsia was recorded in the same column. We found it extremely challenging to be specific between the two complications. We, therefore, combined preeclampsia and eclampsia. We also had very few cases of maternal sepsis as such we combined APH and sepsis during analysis of obstetric complications. This should be interpreted with caution as it might differ with findings from other studies where these two complications were analyzed separately. Research has shown that maternal predictors also can vary based on birthweight categorizations. Data for infants, however, were not classified as small-for-gestational-age (SGA) in the current study. We did not include newborns with other adverse neonatal outcomes such as macerated and fresh stillbirths. This may limit our inferences and interpretation of some results, especially in the comparisons between two high risk groups such as adolescent girls and women of advanced maternal age or between different obstetric complications. However, our results are less likely to be due to chance as it had a lot of study participants, hence, they are reliable and more generalizable.

## Conclusion

This study observed a high prevalence of LBW at Rev. John Chilembwe Hospital in Phalombe district in Malawi. We also established that prematurity, preeclampsia, including eclampsia, antepartum hemorrhage, including sepsis were predictors of LBW at this hospital. The study found significant differences in median birthweights of newborns among different groups of

maternal factors (age groups, parity, obstetric complications and gestational age of pregnancy). Our study, however, showed no statistically significant differences in median birthweights between newborns of HIV+ and HIV- women as well as among newborns delivered through different delivery modes. We recommend that interventions that are aimed at preventing prematurity, preeclampsia and eclampsia, antepartum hemorrhage and sepsis should be prioritized to improve birthweight outcomes at Rev. John Chilembwe Hospital in Phalombe district in Malawi.

## Supporting information

**S1 Dataset.**
(XLS)

## Acknowledgments

Our gratitude goes to the District Health Management Team (DHMT) of Phalombe District Health Office (DHO) for granting us permission to conduct this study at Rev. John Chilembwe Hospital in Phalombe district.

## Author Contributions

**Conceptualization:** Dumisani Mfipa.

**Data curation:** Dumisani Mfipa.

**Formal analysis:** Dumisani Mfipa.

**Investigation:** Dumisani Mfipa.

**Methodology:** Dumisani Mfipa.

**Writing – original draft:** Dumisani Mfipa.

**Writing – review & editing:** Dumisani Mfipa, Precious L. Hajison, Felistas Mpachika-Mfipa.

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
