## [Decision Letter · Decision Letter 0]

10 Oct 2023

PONE-D-23-27598Comparisons of newborn birthweights with maternal factors at Phalombe District Hospital, Malawi: a retrospective record reviewPLOS ONE

Dear Dr. Mfipa,

Thank you for submitting your manuscript to PLOS ONE. After careful consideration, we feel that it has merit but does not fully meet PLOS ONE’s publication criteria as it currently stands. Therefore, we invite you to submit a revised version of the manuscript that addresses the points raised during the review process.

This manuscript describes a retrospective analysis of healthcare data related to birthweight from Malawi. The main dependent variable in the analysis, birthweight, was analysed as a continuous variable converted into ranks because of its skewed distribution. The two reviewers differ slightly in their considered opinion of the quality of the manuscript. In the fourth criteria for publication in this journal there is a stated need for conclusions that are presented in an appropriate fashion and are supported by the data. The concerns highlighted by reviewer 2 in particular suggest to me that this criterion has not so far been met. However, I believe that the authors have the possibility of being able to meet it if they robustly deal with the various points raised, especially the points about the need to define the various adverse conditions of pregnancy, why some of pregnancy variables were analysed in binary form (yes/no), where is the effect of gestation in the analyses (when this has a major association with birth weight) and why stillbirths were excluded from the study when they are related to low birth weight. From reviewer 1, I agree that it is crucial that low birth weight (< 2.5 kg) is analysed as a key variable and agree that it would be more meaningful to present equivalent birth weights rather than birth weight ranks (which don't mean anything to the reader). If the authors feel that they can robustly deal with all the points raised by the reviewers, especially the ones that I have highlighted, then I would be prepared to reconsider the manuscript. 

We look forward to receiving your revised manuscript.

Kind regards,

Clive J. Petry, PhD

Academic Editor

PLOS ONE

5. We are unable to open your Supporting Information file [Dataset.sav]. Please kindly revise as necessary and re-upload.

Reviewers' comments:

Reviewer's Responses to Questions

**Comments to the Author**

1. Is the manuscript technically sound, and do the data support the conclusions?

Reviewer #1: Partly

Reviewer #2: No

2. Has the statistical analysis been performed appropriately and rigorously? 

Reviewer #1: No

Reviewer #2: No

3. Have the authors made all data underlying the findings in their manuscript fully available?

Reviewer #1: Yes

Reviewer #2: Yes

4. Is the manuscript presented in an intelligible fashion and written in standard English?

Reviewer #1: Yes

Reviewer #2: Yes

5. Review Comments to the Author

Reviewer #1: The manuscript entitled “Comparisons of newborn birthweights with maternal factors at Phalombe District Hospital, Malawi: a retrospective record review” compares the birthweights of infants born to different maternal groups in Malawi, Africa. Using data from the birth registry of Phalombe District Hospital, the authors examine the impact of four maternal variables, age, HIV status, parity, and obstetric complications, on birthweight. This study holds significance as it highlights the factors that can have an impact on birthweight in Malawi. Some recommendations to enhance the manuscript are outlined below:

1. The background is focused on low birth weight (LBW) in infants and maternal risk factors for LBW; however, comparisons have been made on median birthweight in different maternal groups. It would be good to check if the prevalence of LBW was different among the maternal categories presented. Also, rather than focusing solely on LBW, explore these differences using data on infants classified as small-for-gestational-age: SGA (birthweight below the 10th percentile for gestational age). For instance, a newborn with a birthweight of 2,400 grams could fall into the SGA, AGA, or LGA categories, and the associated outcomes would vary depending on the specific category the newborn belongs to. Research has shown that maternal predictors also can vary based on birthweight categorisations. I think additional analyses that integrate data on birthweight standardised for gestational age would enhance the interpretability of the findings.

2. The authors have used non-parametric tests as the independent variables did not follow a normal distribution; however, there is no mention of whether the outcome variable (birthweight) was normally distributed. Did the authors consider using multivariable logistic regression analysis, which can be used to assess the relationship between several independent variables and outcome variables such as LBW or SGA? It also allows the assessment of independent relationships while adjusting for potential confounders. Logistic regression assumes that the log odds (logit) of the dependent variable is a linear combination of the independent variables. This means that the relationship between the independent variables and the log odds of the outcome variable is linear, but it does not require the independent variables themselves to be normally distributed.

3. Only mean ranks of birthweight are presented in the results. Suggest providing the birthweights representing those mean ranks for meaningful results.

4. Are there any other studies which support the finding that “there were no statistically significant differences in mean ranks of newborn birthweights born to HIV positive and HIV negative women”? I would take caution on this result. Could this have resulted due to confounding factors, importantly gestation length?

5. Did the authors have data on other important maternal risk factors for LBW and SGA, such as smoking, drug abuse, being underweight, etc. There should be at least a discussion on those factors.

Reviewer #2: Thank you for the opportunity to review this paper. The aim was to compared newborn birthweights with maternal risk factors at a hospital in Malawi.

It was a retrospective study of 1308 women who gave birth over a 6 month period.

How accurate was the recording of birthweight and the different maternal conditions especially pre eclampsia, sepsis and APH. There are no definitions provided or any sense of magnitude. Would one episode of APH at 24 weeks classify as APH in a woman who have birth at term? What magnitude of sepsis was used, what definition and when did the sepsis occur (I presume during pregnancy but this is not clear). Why is (pre) eclampsia written with the brackets? I have never seen this before.

These conditions were then coded dichotomously – absent or present. This feels very blunt. How can we be sure that this is sensitive enough?

Was the use of ART considered in the HIV positive women and again, the range of health care status in women with HIV. Just being positive feels a bit blunt.

Was the mode of birth considered? What is the rate of induction of labour or elective caesarean section in this hospital?

How many women/babies had missing data?

I was surprised that stillbirths were excluded. Surely stillbirth is an example of an impact of low birth weight?

How was gestation controlled for in the study? I cannot see any information about gestation or how this was considered in the analysis. This seems to be a major flaw and an explanation that is not discussed.

6. PLOS authors have the option to publish the peer review history of their article (what does this mean?). If published, this will include your full peer review and any attached files.

Reviewer #1: No

Reviewer #2: No

---

## [Author Response · Author response to Decision Letter 0]

13 Dec 2023

Reviewer 1: The manuscript entitled “Comparisons of newborn birthweights with maternal factors at Phalombe District Hospital, Malawi: a retrospective record review” compares the birthweights of infants born to different maternal groups in Malawi, Africa. Using data from the birth registry of Phalombe District Hospital, the authors examine the impact of four maternal variables, age, HIV status, parity, and obstetric complications, on birthweight. This study holds significance as it highlights the factors that can have an impact on birthweight in Malawi. Some recommendations to enhance the manuscript are outlined below: 

Response to Reviewer 1: Thank you. This title has now been revised to “Predictors of low birthweight and comparisons of newborn birthweights with maternal factors at Phalombe District Hospital, Malawi: a retrospective record review.” (See Lines 1 to 3)

Reviewer 1: The background is focused on low birth weight (LBW) in infants and maternal risk factors for LBW; however, comparisons have been made on median birthweight in different maternal groups. It would be good to check if the prevalence of LBW was different among the maternal categories presented. Also, rather than focusing solely on LBW, explore these differences using data on infants classified as small-for-gestational-age: SGA (birthweight below the 10th percentile for gestational age). For instance, a newborn with a birthweight of 2,400 grams could fall into the SGA, AGA, or LGA categories, and the associated outcomes would vary depending on the specific category the newborn belongs to. Research has shown that maternal predictors also can vary based on birthweight categorisations. I think additional analyses that integrate data on birthweight standardised for gestational age would enhance the interpretability of the findings. 

Response to Reviewer 1: Thank you. We have done the comparisons about the prevalence of LBW among different maternal categories (See Table 1, Lines 219 to 220). This study dwelt on LBW as such we have included 'lack of classification of SGA' as a limitation and calling on future research to consider addressing this gap (See Lines 548 to 549).

Reviewer 1: The authors have used non-parametric tests as the independent variables did not follow a normal distribution; however, there is no mention of whether the outcome variable (birthweight) was normally distributed. Did the authors consider using multivariable logistic regression analysis, which can be used to assess the relationship between several independent variables and outcome variables such as LBW or SGA? It also allows the assessment of independent relationships while adjusting for potential confounders. Logistic regression assumes that the log odds (logit) of the dependent variable is a linear combination of the independent variables. This means that the relationship between the independent variables and the log odds of the outcome variable is linear, but it does not require the independent variables themselves to be normally distributed. 

Response to Reviewer 1: Thank you. We have included a statement to show that the outcome variable was skewed. See Table 3, Lines 249 to 250. We have performed logistic regression analyses (See Table 4, Lines 329 to 330).

Reviewer 1: Only mean ranks of birthweight are presented in the results. Suggest providing the birthweights representing those mean ranks for meaningful results. 

Response to Reviewer 1: Thank you. We have replaced mean ranks with median birthweights which are more meaningful. We also changed the method of data analysis. We have now used Median Test as the data contains some outliers.

Reviewer 1: Are there any other studies which support the finding that “there were no statistically significant differences in mean ranks of newborn birthweights born to HIV positive and HIV negative women”? I would take caution on this result. Could this have resulted due to confounding factors, importantly gestation length? 

Response to Reviewer 1: Thank you. It is important to take note that we compared birthweights with maternal factors using one-way Median test. Each test was independent of the other variables as such these comparisons did not control for any confounding factors. We have, however, taken note of this input and we have included gestation length (in weeks) which has been categorized into term (37 – 41.6 weeks), preterm (≤36.6 weeks) and postterm (≥42 weeks) (See Lines 142 to 144). However, in our logistic regression analysis, this has been taken care of where gestation age and other variables were adjusted for. After performing logistic regression analysis still shows that being HIV+ is not statistically associated with LBW deliveries at univariate level (OR = 1.01, 95% CI 0.61 - 1.67, p = .979) (See Table 4, Lines 329 to 330). 

For your information, other studies have found similar results. For example, a study by Habib et al (Habib NA, Daltveit AK, Bergsjø P, Shao J, Oneko O & Lie RT. Maternal HIV status and pregnancy outcomes in northeastern Tanzania: a registry-based study. BJOG. (2008) 115 (5), 616 – 624. https://doi.org/10.1111/j.1471-0528.2008.01672.x) found that treated HIV-infected women had a risk similar to that of the HIV-uninfected women for all outcomes (perinatal mortality, prematurity, SGA, birthweight), except for low Apgar score. All HIV+ women in our study were on ART treatment (as indicated in the maternity registers). On the contrary, a study by Wedi et al. found that maternal HIV infection in women who have not received antiretroviral therapy is associated with preterm birth, low birthweight, small for gestational age, and stillbirth, especially in sub-Saharan Africa. No association was identified between maternal HIV infection and very preterm birth, very small for gestational age, very low birthweight, miscarriage, or neonatal death, although few data were available for these outcomes. Correction for confounders did not affect the significance of these findings. (Wedi COO, Kirtley S, Hopewell, S, Corrigan, R, Kennedy SH, Hemelaar J. Perinatal outcomes associated with maternal HIV infection: a systematic review and meta-analysis. (2016) 3 (1), E33-E48. The Lancet HIV. https://doi.org/10.1016/S2352-3018(15)00207-6). Wedi et al 

Reviewer 1: Did the authors have data on other important maternal risk factors for LBW and SGA, such as smoking, drug abuse, being underweight, etc. There should be at least a discussion on those factors. 

Response to Reviewer 1: Thank you. Data on smoking, drug abuse and being underweight, etc are not captured in the maternity registers in Malawi as such we did not have such data at our disposal. We have, however, included this as study limitation (See Lines 530 to 532).

Reviewer 2: Thank you for the opportunity to review this paper. The aim was to compared newborn birthweights with maternal risk factors at a hospital in Malawi. It was a retrospective study of 1308 women who gave birth over a 6 month period. 

Response to Reviewer 2: Thank you.

Reviewer 2: How accurate was the recording of birthweight and the different maternal conditions especially pre eclampsia, sepsis and APH. There are no definitions provided or any sense of magnitude. Would one episode of APH at 24 weeks classify as APH in a woman who have birth at term? What magnitude of sepsis was used, what definition and when did the sepsis occur (I presume during pregnancy but this is not clear). Why is (pre) eclampsia written with the brackets? I have never seen this before. 

Response to Reviewer 2: Thank you. The recording of birthweight was accurate as this was done soon after delivery before the baby was taken to postnatal ward. In additional, all the weights were recorded in grams and those newborns who had missing birth weight data were excluded from the analysis. For the maternal conditions (pre-eclampsia, sepsis and APH), the register that we extracted data from already had these conditions listed and any woman who presented with such a condition during labour and delivery, was recorded accordingly. For the definitions of the maternal conditions, this has been noted and operational definitions have been provided (See Lines 128 to 144).

(Pre) Eclampsia has been written with brackets because this is how the register from where the data was extracted has been designed. This means that any woman with Pre-eclampsia or with Eclampsia is recorded in the same category.

We have included the magnitude of this problem in the introduction (See Lines 69 to 71). You will also find the findings on the same in results section. We defined obstetric complications as pregnant women having (pre)-eclampsia, antepartum hemorrhage and sepsis. These were selected due to their effect on LBW from previous studies. We did not want to include complications that have no scientific backing as factors associated with LBW. 

As you will have noted from the study, we collected data retrospectively from maternity registers. This made it impossible for us to know when the APH episodes started occurring during pregnancy as the registers do not specify this. One thing for sure is that episodes occurring during birth time were recorded.

Reviewer 2: These conditions were then coded dichotomously – absent or present. This feels very blunt. How can we be sure that this is sensitive enough? 

Response to Reviewer 2: Thank you. We have revised the way we coded obstetric complications. This has now been categorized and coded as below: women with pre-eclampsia, including those with eclampsia (1), women with antepartum hemorrhage, including those with sepsis (2) and control group [refers to women with no pre-eclampsia/eclampsia, antepartum hemorrhage and sepsis] (3). See Lines 139 – 141.

NOTE: 1 woman had sepsis and delivered a LBW baby. This has been combined with APH. Secondly, the maternal registers just have column labelled '(pre)-eclampsia.' This is a column that captures women with pre-eclampsia and eclampsia. Health workers do not specify if what they are recording is either pre-eclampsia or eclampsia. That is why we have coded it the way we have. 

Reviewer 2: Was the use of ART considered in the HIV positive women and again, the range of health care status in women with HIV. Just being positive feels a bit blunt. 

Response to Reviewer 2: Thank you. All HIV+ women in our study were on ART. We have added a statement to indicate that (See Lines 471 to 474).

Reviewer 2: Was the mode of birth considered? What is the rate of induction of labour or elective caesarean section in this hospital? 

Response to Reviewer 2: Thank you. We have included mode of delivery and it has been categorized as normal deliveries, caesarean section deliveries and assisted vaginal deliveries (See Lines 131 to 135).

Reviewer 2: How many women/babies had missing data? 

Response to Reviewer 2: Thank you. We have included the statement below under the results section.

Eighty-seven (87) women did not meet the inclusion criteria as 69 had their gestation weeks either unrecorded or unclear, 13 had their parity unclear, 4 had their ages unrecorded and 1 had her HIV status unrecorded. One hundred eighty-eight (188) newborns were not included as 47 had birthweights unrecorded, 104 were twins and 37 were stillbirths (macerated stillbirths = 14 and fresh stillbirths = 23) (See Lines 200 to 204).

Reviewer 2: I was surprised that stillbirths were excluded. Surely stillbirth is an example of an impact of low birth weight? 

Response to Reviewer 2: Thank you. We excluded stillbirths to be in line with the definition of incidence of LBW. UNICEF and WHO define incidence of LBW in a population as the percentage of live births that weigh less than 2,500g out of the total of live births during the same time period (United Nations Children’s Fund and World Health Organization. Low birthweight: Country, regional and global estimates. New York: UNICEF; 2004) (See Lines 112 to 114). 

Reviewer 2: How was gestation controlled for in the study? I cannot see any information about gestation or how this was considered in the analysis. This seems to be a major flaw and an explanation that is not discussed. 

Response to Reviewer 2: Thank you. Please note that gestation has been controlled for in our study when conducting multivariate regression analysis (See Table 4, Lines 329 to 330).

---

## [Decision Letter · Decision Letter 1]

20 Mar 2024

PONE-D-23-27598R1Predictors of low birthweight and comparisons of newborn birthweights with maternal factors at Phalombe District Hospital, Malawi: a retrospective record reviewPLOS ONE

Dear Dr. Mfipa,

Thank you for submitting your manuscript to PLOS ONE. After careful consideration, we feel that it has merit but does not fully meet PLOS ONE’s publication criteria as it currently stands. Therefore, we invite you to submit a revised version of the manuscript that addresses the points raised during the review process.

We look forward to receiving your revised manuscript.

Kind regards,

George Kuryan

Academic Editor

PLOS ONE

Journal Requirements:

Reviewers' comments:

Reviewer's Responses to Questions

**Comments to the Author**

1. If the authors have adequately addressed your comments raised in a previous round of review and you feel that this manuscript is now acceptable for publication, you may indicate that here to bypass the “Comments to the Author” section, enter your conflict of interest statement in the “Confidential to Editor” section, and submit your "Accept" recommendation.

Reviewer #1: All comments have been addressed

Reviewer #3: (No Response)

2. Is the manuscript technically sound, and do the data support the conclusions?

Reviewer #1: Yes

Reviewer #3: Yes

3. Has the statistical analysis been performed appropriately and rigorously? 

Reviewer #1: Yes

Reviewer #3: Yes

4. Have the authors made all data underlying the findings in their manuscript fully available?

Reviewer #1: Yes

Reviewer #3: Yes

5. Is the manuscript presented in an intelligible fashion and written in standard English?

Reviewer #1: Yes

Reviewer #3: Yes

6. Review Comments to the Author

Reviewer #1: Thank you for the revisions on the manuscript. I appreciate the effort you put into addressing my feedback. I have no further comments.

Reviewer #3: Thank you for the opportunity to review the manuscript. The manuscript aims to identify the predictors of low birth weight based on retrospective data from birth registers at Phalombe district hospital. The research question and methodology are clear, and the discussion is detailed. However, I have a few questions and minor recommendations.

1. Is Table 2 necessary? Since it is a single row and explained in the text, the table seems redundant.

2. In the introduction (lines 69-71) it is stated that there were 12% LBW deliveries from 10th October to 31st December 2022 and 9% LBW deliveries from 01st January to 31st March. Is there a reason the final sample in this study has a much higher proportion of LBW?

3. The authors have clarified the reason for not including the other adverse neonatal outcomes such as stillbirths. However, this is a limitation in the interpretation of some results. Especially in the comparisons between two high risk groups such as 'adolescent girls' and 'women of advanced maternal age' (lines 379-380) or between different obstetric complications, the non-inclusion of other adverse neonatal outcomes limits the inferences.

4. The discussion and conclusion can be more concise and nuanced. Maternal age is emphasized as an important variable in the discussion without any attempt to explain the why Maternal age is not significant in the multivariate analysis. Maternal age may also be related to other social and economic variables which may affect obstetric complications with complex, intersectional relationships. Hence, more data is needed to identify the interventions in this area. Some recommendations are too generalized and not derived from the study results (lines 431, 581-583).

7. PLOS authors have the option to publish the peer review history of their article (what does this mean?). If published, this will include your full peer review and any attached files.

Reviewer #1: No

Reviewer #3: No

---

## [Author Response · Author response to Decision Letter 1]

2 May 2024

Comments by Reviewer 1: Thank you for the revisions on the manuscript. I appreciate the effort you put into addressing my feedback. I have no further comments.

Authors’ Response: Thank you.

Comments by Reviewer 3: Thank you for the opportunity to review the manuscript. The manuscript aims to identify the predictors of low birth weight based on retrospective data from birth registers at Phalombe district hospital. The research question and methodology are clear, and the discussion is detailed. However, I have a few questions and minor recommendations.

Authors’ Response: Thank you.

Comments by Reviewer 3: Is Table 2 necessary? Since it is a single row and explained in the text, the table seems redundant.

Authors’ Response: Thank you. Table 2 has been removed while the text explaining it has been retained (See Page 13, Lines 220 – 222 in the track change version).

Comments by Reviewer 3: In the introduction (lines 69-71) it is stated that there were 12% LBW deliveries from 10th October to 31st December 2022 and 9% LBW deliveries from 01st January to 31st March. Is there a reason the final sample in this study has a much higher proportion of LBW?

Authors’ Response: Thank you. We observed a higher prevalence of LBW for the study period of October, 2022 to March, 2023 than 12% and 9% for October to December, 2022 and January to March, 2023, respectively, as reported by the hospital. This difference came about due to data cleaning challenges among midwives. In few instances, midwives had failed to indicate that babies were LBW even though they had entered the newborn birthweights as less than 2,500g. This led to incorrect LBW page summary calculations in the maternity registers, which resulted in lower proportion of LBW. With reference to WHO definition of LBW, we rectified such errors by correctly categorizing LBW as those newborns weighing less than 2,500g since the data on birthweight was available to us in the maternity registers. Poor data quality on LBW was also noted in the Malawi Multiple Indicator Cluster Survey 2019-2020 report, and it recommended cautious use of this data as it was considered an under estimation of the true prevalence (National Statistical Office, 2021) (See Page 20, Lines 352 – 360 in the track change version)

Comments by Reviewer 3: The authors have clarified the reason for not including the other adverse neonatal outcomes such as stillbirths. However, this is a limitation in the interpretation of some results. Especially in the comparisons between two high risk groups such as 'adolescent girls' and 'women of advanced maternal age' (lines 379-380) or between different obstetric complications, the non-inclusion of other adverse neonatal outcomes limits the inferences.

Authors’ Response: Thank you. This has been noted but it was necessary for us to stick to the operational definition of incidence of LBW in a population as the percentage of live births that weigh less than 2,500g out of the total of live births during the same period. As a result, we had to exclude the stillbirths. However, we have included this as our study limitation (See Page 26, Lines 499 to 503 in the track change version).

Comments by Reviewer 3: The discussion and conclusion can be more concise and nuanced. Maternal age is emphasized as an important variable in the discussion without any attempt to explain the why Maternal age is not significant in the multivariate analysis. Maternal age may also be related to other social and economic variables which may affect obstetric complications with complex, intersectional relationships. Hence, more data is needed to identify the interventions in this area. Some recommendations are too generalized and not derived from the study results (lines 431, 581-583).

Authors’ Response: Thank you. We have revised the conclusion section to make it more concise and nuanced (See Page 27 Lines 506 to 514 in the track change version). We have removed the sub-headings in the discussion section. Generally, its content has been left as it was so that we do not dilute it. We have also explained why maternal age may not have been significant in the multivariable analysis (See Pages 18 and 19, Lines 316 to 334 in the track change version). This statement has been inserted in the discussion section under maternal age. We have removed the recommendations that were deemed too generalized and not derived from the study results.

---

## [Decision Letter · Decision Letter 2]

16 Aug 2024

Predictors of low birthweight and comparisons of newborn birthweights among different groups of maternal factors at Rev. John Chilembwe Hospital in Phalombe district, Malawi: a retrospective record review

PONE-D-23-27598R2

Dear Dumisani Mfipa,

We’re pleased to inform you that your manuscript has been judged scientifically suitable for publication and will be formally accepted for publication once it meets all outstanding technical requirements.

Kind regards,

Sidrah Nausheen, FCPS

Academic Editor

PLOS ONE

Additional Editor Comments (optional):

Reviewers' comments:

Reviewer's Responses to Questions

**Comments to the Author**

1. If the authors have adequately addressed your comments raised in a previous round of review and you feel that this manuscript is now acceptable for publication, you may indicate that here to bypass the “Comments to the Author” section, enter your conflict of interest statement in the “Confidential to Editor” section, and submit your "Accept" recommendation.

Reviewer #1: All comments have been addressed

Reviewer #3: All comments have been addressed

2. Is the manuscript technically sound, and do the data support the conclusions?

Reviewer #1: Yes

Reviewer #3: Yes

3. Has the statistical analysis been performed appropriately and rigorously? 

Reviewer #1: Yes

Reviewer #3: Yes

4. Have the authors made all data underlying the findings in their manuscript fully available?

Reviewer #1: Yes

Reviewer #3: Yes

5. Is the manuscript presented in an intelligible fashion and written in standard English?

Reviewer #1: Yes

Reviewer #3: Yes

6. Review Comments to the Author

Reviewer #1: (No Response)

Reviewer #3: Thank you for revisions. I have no further questions. Moving some details in the statistical analysis such as the table on normality tests will improve the readability of the manuscript.

7. PLOS authors have the option to publish the peer review history of their article (what does this mean?). If published, this will include your full peer review and any attached files.

Reviewer #1: No

Reviewer #3: No

---

## [Editor Report · Acceptance letter]

21 Aug 2024

PONE-D-23-27598R2 

PLOS ONE

Dear Dr. Mfipa, 

I'm pleased to inform you that your manuscript has been deemed suitable for publication in PLOS ONE. Congratulations! Your manuscript is now being handed over to our production team.

Kind regards, 

on behalf of

Dr. Sidrah Nausheen 

Academic Editor

PLOS ONE